# Fibre Bragg Grating Based Interface Pressure Sensor for Compression Therapy

**DOI:** 10.3390/s22051798

**Published:** 2022-02-24

**Authors:** James A. Bradbury, Qimei Zhang, Francisco U. Hernandez Ledezma, Ricardo Correia, Serhiy Korposh, Barrie R. Hayes-Gill, Ferdinand Tamoué, Alison Parnham, Simon A. McMaster, Stephen P. Morgan

**Affiliations:** 1Optics and Photonics Group, Faculty of Engineering, University of Nottingham, Nottingham NG7 2RD, UK; james.bradbury@nottingham.ac.uk (J.A.B.); ricardo.goncalvescorreia@nottingham.ac.uk (R.C.); s.korposh@nottingham.ac.uk (S.K.); barrie.hayes-gill@nottingham.ac.uk (B.R.H.-G.); 2Department of Engineering, School of Science and Technology, Nottingham Trent University Nottingham, Nottingham NG1 4FQ, UK; qimei.zhang@ntu.ac.uk; 3Footfalls and Heartbeats (UK) Limited, 10 Castle Quay, Castle Boulevard, Nottingham NG7 1FW, UK; ulises@footfallsandheartbeats.com (F.U.H.L.); simon@footfallsandheartbeats.com (S.A.M.); 4KOB GmbH, Lauterstraße 50, 67752 Wolfstein, Germany; ferdinand.tamoue@kob.de; 5School of Health Sciences, University of Nottingham, Nottingham NG7 2RD, UK; alison.parnham@nottingham.ac.uk

**Keywords:** optical fibre sensor, compression therapy, venous leg ulcer, sub-bandage pressure sensor, Fibre Bragg Grating

## Abstract

Compression therapy is widely used as the gold standard for management of chronic venous insufficiency and venous leg ulcers, and the amount of pressure applied during the compression therapy is crucial in supporting healing. A fibre optic pressure sensor using Fibre Bragg Gratings (FBGs) is developed in this paper to measure sub-bandage pressure whilst removing cross-sensitivity due to strain in the fibre and temperature. The interface pressure is measured by an FBG encapsulated in a polymer and housed in a textile to minimise discomfort for the patient. The repeatability of a manual fabrication process is investigated by fabricating and calibrating ten sensors. A customized calibration setup consisting of a programmable translation stage and a weighing scale gives sensitivities in the range 0.4–1.5 pm/mmHg (2.6–11.3 pm/kPa). An alternative calibration method using a rigid plastic cylinder and a blood pressure cuff is also demonstrated. Investigations are performed with the sensor under a compression bandage on a phantom leg to test the response of the sensor to changing pressures in static situations. Measurements are taken on a human subject to demonstrate changes in interface pressure under a compression bandage during motion to mimic a clinical application. These results are compared to the current gold standard medical sensor using a Bland–Altman analysis, with a median bias ranging from −4.6 to −20.4 mmHg, upper limit of agreement (LOA) from −13.5 to 2.7 mmHg and lower LOA from −32.4 to −7.7 mmHg. The sensor has the potential to be used as a training tool for nurses and can be left in situ to monitor bandage pressure during compression therapy.

## 1. Introduction

Compression therapy is the application of pressure to the body, usually to the limb, using bandages, hosiery, or other devices. It is established as the treatment of choice for venous diseases such as chronic venous insufficiency and venous leg ulcers (VLUs), aiming to reduce and correct the symptoms of valvular incompetence and venous hypertension [1,2]. In addition, compression therapy is also used in the treatment and prevention of lymphoedema and burn scars [3]. 

Up to 3% of the global adult population will be affected by VLUs [4,5], creating a financial burden on health services around the world. A study in the United Kingdom analysed 11 years’ worth of data (2007–2017) and found an annual national cost of over £2 billion for the treatment of VLUs [5]. A study in the Sichuan province in China found a point prevalence for VLUs of 0.28 per 1000 population [6], and this is likely an underestimate as it only considers hospital in-patients. In Australia, models suggest that over 300,000 people will be affected by VLUs, with the total cost over 5 years of treatment reaching between 1.2 and 2.8 billion AUD [7]. A study in the US comparing the costs of VLU treatment found annual values of $6391 for people with Medicare (US national healthcare insurance) and $7086 for privately insured patients, producing an annual US payer burden of $14.9 billion [8].

The amount of pressure applied during compression therapy is crucial in supporting wound healing. Insufficient pressure impairs the efficacy, while excessive compression can interrupt circulation and cause aggravated tissue damage. For example, tibial crest and medial malleolus are positions that are particularly vulnerable to pressure damage [9]. The compression applied is influenced by a number of factors including the physical structure and viscoelastic properties of the bandage fabric; the size and shape of the limb; and the skill and technique of the healthcare professional (HCP) or user [10]. Furthermore, the sub-bandage pressure is likely to change over time as the tension in the fabric decreases as a result of the movement of the patient’s limb [11]. Applying a bandage is a skill that must be learned and practiced regularly; however, even an experienced HCP will find it challenging to maintain consistent quality in compression bandaging [12]. These complications with the treatment mean that significant practitioner time is spent applying and managing compression bandages, with some estimates from the UK suggesting up to 65% of community nurses’ time is spent treating VLUs [13]. The difference in cost for managing an unhealed VLU or using a sub-optimal treatment pathway is estimated to be up to 10 times higher than successful treatment [4,14].

The ability to quantify the levels of compression can help to educate new practitioners or HCPs, benefit development of new compression bandages, and enable a fundamental understanding of compression therapy.

A modified Laplace’s law was proposed to predict the pressure ((P) in mmHg) applied by a compression bandage and is expressed as [15]:P = (TN × 4620)/C × W, (1)
where T is bandage tension in kgf, N is the number of layers applied, C is the circumference of the limb and W is the width of the bandage, both in cm. The constant 4620 is used to convert the unit of the pressure P into mmHg, which is widely used in clinical applications and so is maintained in this paper for consistency (1 KPa = 7.5 mmHg). Tamoue et al. demonstrated that it is possible to use the equation to predict the pressure developed by a compression bandage on a limb of known circumference assuming the leg has a circular cross section [16]. However, the pressure calculated is the average and there can be significant differences at different positions on the limb. For example, the pressure could range from near zero in a concavity to a value an order of magnitude greater than the calculated mean when measured over the tibial crest [10]. Disagreements between the predictions of Laplace’s law and the measurement devices used for sub-bandage pressures have also led to a questioning of whether this model is suitable [17], though others argue that this is an issue with the devices rather than the model [10].

Most commercially available pressure sensors are deemed unsuitable to be used to measure pressure in compression therapy due to the inability to measure low pressure levels (<70 mmHg (9.33 kPa) for compression therapy) and excessive thickness and dimensions [18]. Several air-filled devices were developed specifically to measure the sub-bandage pressure, such as the Kikuhime (Meditrade, Soro, Denmark), SIGaT-tester (Ganzoni-Sigvaris, St. Gallen, Switzerland), and PicoPress (Microlab, Padua, Italy) [19]. These devices have a circular plastic bladder which is filled with air. During the measurement, air is injected into the bladder through a syringe and the air pressure is measured with a manometer. Such devices are thin, flexible and can provide continuous registration of pressure over time. However, it has been demonstrated that the accuracy of the air-based pressure sensors is greatly reduced on curved surfaces, in free air or beneath bandages [20]. In addition, the plastic bladder must be connected to the manometer using rigid plastic tubes, which can cause damage to people with fragile skin during compression therapy.

There have been attempts to improve the comfort for the patient by using smaller and thinner piezoresistive sensors; for example, the FlexiForce (Tekscan Inc., Boston, MA, USA), but commercially available piezoresistive devices have yet to recreate the accuracy and reliability of the air-filled sensors described above [21]. There has been some success on flatter areas of the body [22], suggesting that there may still be a clinical application for piezoresistive devices.

Fibre Bragg Grating (FBG) based sensors have advantages including a small size, a light weight, biocompatibility, chemical inertness, and immunity to electromagnetic interference. FBG-based sensors have found applications in aerospace, biochemistry, structure monitoring in civil engineering, and biomedical devices [23]. An FBG-based optical fibre sensor to measure the sub-bandage pressure has also been developed [11,24,25]. The sensor is made of two FBG arrays entwined in a double helix form which is provided thermal and axial strain insensitivity. Using multiple FBG arrays, the sensor can measure the sub-bandage resting pressure profiles across the full length of a human leg. The pressure variation during a patient’s movement was also recorded and it has provided an insight into sub bandage pressure distribution. Given the promising results obtained from such sensors in bandage/tissue interface pressure measurements, it is therefore of interest to explore the performance of different configurations of optical fibre pressure sensors. 

In this paper, a new configuration of an FBG-based optical sensor is developed for application in compression therapy. The sensitivity of the FBG to pressure is increased by embedding the FBG inside a polymer. The ability of the pressure sensor to measure the absolute pressures at the B1 point [26] (the transition of the muscular part of the medial gastrocnemius muscle into the tendinous part) on a healthy human leg whist applying compression bandages is verified. The sensing system is calibrated and validated under laboratory conditions and is then tested on a healthy human subject for a range of motions. The results are compared with those from the current state of the art device (PicoPress). The sensor has the potential to be used as a training tool for nurses and, as the sensor is thin, lightweight, and low cost, it can be left in situ to frequently monitor bandage pressure to support the treatment of VLUs and lymphoedema. 

## 2. Materials and Methods

In this section, the theoretical principles of an FBG sensor, the design choices made to create an FBG sensor capable of isolating a pressure measurement and the techniques used to calibrate the sensor are explained. The methodology behind the two methods used to validate the response of the sensor in non-clinical scenarios, and the procedure used to mimic the application of the sensor to a clinical measurement, are then described. 

### 2.1. Principles of the Pressure Sensor

An FBG consists of a periodic modulation of the refractive index of an optical fibre core. It acts as a narrow band filter that only reflects light at the Bragg wavelength (λ_B_) of
λ_B_ = 2 × n × Λ,(2)
while transmitting light at other wavelengths. In Equation (2), n is the effective refractive index of the core and Λ is the grating period. Changes in force applied to the FBG will affect both n and Λ; therefore, the Bragg wavelength λ_B_ is a function of force. Force and pressure are linked by considering the area over which the force is applied, therefore converting the FBG from a force sensor to a pressure sensor is a straightforward calculation once the effective area of the sensor has been identified. For the rest of this section, we will refer to pressure instead of force.

The effective refractive index and grating period, and so the Bragg wavelength, are also affected by temperature and strain as shown in Equation (3) [27]: with P_e_ being the photo-elastic constant, ε the isotropic strain, α the thermal expansion coefficient, ξ the thermo-optic coefficient and ΔT the temperature variance:∆λ_B_ = λ_B_ [(1 − P_e_) × ε + ((1 − P_e_)× α + ξ) × ∆T],(3)

Meaning the Bragg wavelength shift can be summarised as [28,29]
∆λ_B_ = K_T_ × ∆T + K_σz_ × σz + K_P_ × P,(4)
where K_T_ is the sensitivity coefficient for temperature, K_σz_ is the sensitivity coefficient for axial strain and K_P_ is the sensitivity coefficient for transversely loaded pressure; with ∆T, σz, and P being the temperature change, axial strain, and pressure, respectively.

Therefore, during calibration, it is necessary to compensate for temperature and strain cross-sensitivity. Some examples of FBG sensors use a diaphragm system over the fibre to apply the pressure. This system has the advantage of being able to use the thermal properties of the diaphragm to compensate for temperature effects on the sensor [30]. However, the use of a diaphragm often involves a rigid structure, forming part of the sensor. Due to the medical application being investigated here, where the sensor is used near the site of a wound where the skin of the patient is delicate, it is advantageous to avoid using rigid structures.

### 2.2. Sensor Preparation

A schematic of the designed optical fibre-based pressure sensor is shown in Figure 1. Three commercially available femtosecond FBGs (FBG_T_, FBG_PST_ and FBG_ST_) with the desired separation and λ_B_ (1545, 1550 and 1555 nm) were bought from FemtoFiberTec (FemtoFiberTec GmbH, Berlin, Germany). The fibre used was a single mode with a 9 μm core, 125 μm cladding and an acrylic coating. 

For a standard FBG (inscribed into a bare optical fibre), the pressure sensitivity is approximately 4 × 10^−4^ pm/mmHg [31] (approximately 0.003 pm/kPa). For high resolution interrogators, the smallest wavelength shift that can be measured is ~0.04 pm which corresponds to a minimum pressure of 100 mmHg (~13 kPa). This is too high for compression therapy in which the pressure range of interest is usually between 0–100 mmHg. To increase the pressure sensitivity, the FBG was therefore encased in a polymer.

The polymer used was Vitralit 1655 (Panacol Adhesives, Frankfurt, Germany), which is a UV curing adhesive recommended by the manufacturer for use in medical devices. Once cured, it has a Young’s modulus of 44 MPa and a tensile strength of 16 MPa. 

Previous work has highlighted the suitability of this material for use in FBG sensing applications. Leal-Junior et al. [32] highlight that coating the FBG with this material “indicates the possibility of obtaining highly sensitive pressure and force sensors”. The paper also highlights that the resin can change the strain response and thermal properties of the FBG. Compensating for the different strain response is discussed in Section 2.3.2. Differences in temperature response are less of a concern as the sub-bandage sensor environment is at a largely constant temperature at equilibrium. 

To set the FBG inside the polymer capsule a small strain is applied to the fibre and the central of the three FBGs is placed into a cavity in a Teflon mould. The fibre runs through the centre of the cavity with the FBG as close to the middle of the capsule as possible. Drops of polymer are then slowly added to excess to prevent a concave meniscus forming. The mould is then cured overnight (minimum 12 h) under UV illumination (λ = 380 nm), and the ‘bulge’ formed by the excess polymer is filed down to create a flat top surface that is parallel to the bottom surface.

When a load is applied, the polymer deforms and transfers the load onto the fibre as an axial strain that changes the period of the FBG over the embedded region [33] and enhances the pressure sensitivity of the FBG. The exact increase in pressure sensitivity depends on the mechanical properties of the polymer but in this case increases it by a factor of around 15 [33]. 

Therefore, as seen in Figure 1, the sensor contains one FBG (FBG_PST_) encapsulated in a polymer to transduce applied pressure into axial strain, and two reference FBGs (FBG_T_ and FBG_ST_) situated outside the polymer to compensate for temperature and strain effects as identified in Equations (3) and (4) and shown in Table 1 below. FBG_T_ only responds to temperature as the distal end can move freely; FBG_ST_ responds to strain and temperature as it is located between the interrogator and the pressure sensors. The presence of two compensation FBGs allows for the isolation of an absolute temperature measurement as well as a pressure measurement.

Although a strain reference is included, it is desirable to minimise strain as much as possible. The optical fibre is therefore located inside a protective tube, Hytrel Furcation Tubing (FT900Y, Thorlabs, Newton, NJ, USA), at either side of the polymer capsule to avoid the fibre being anchored during bandaging, as this can increase the strain on the FBGs. The diameter of the optical fibre is smaller than the inner diameter of the protecting tubing, so the optical fibre can move freely inside the tube. The FBG-based sensor is enclosed inside a textile housing to protect the sensor and to avoid unwanted distortion of the sensor by the bandage. It also avoids direct contact of the bandages with the reference FBG and the fibre, which can also increase the strain. In this study, the Bragg wavelength λ_B_ of each FBG is measured using a SmartScope interrogator (Smartfibres, Bracknell, UK).

### 2.3. Calibration

Previous work carried out within our research group has demonstrated the linear response of the polymer-encapsulated FBG to both load and temperature [34]. Therefore, the calibration work for the design in Figure 1 focused on measuring the response of the sensor whilst it is inside a textile housing to ascertain that the linear response is not affected.

In total, ten sensors were produced and calibrated to investigate the repeatability of the manual fabrication method. The results presented are from one of the sensors, unless explicitly stated to be a mean or median, subject to normality testing.

#### 2.3.1. Loading Arm

The pressure sensor of Figure 1 is calibrated using the setup shown in Figure 2. A metal rod applies a load to the sensor that can be adjusted by changing its vertical position using a programmable translation stage (PI M-403.22S controlled by a PI Mercury Step C-663 (Physik Instrumente, Auburn, Massachusetts, US)). An aluminium plate (Figure 2b) is used to ensure that the load is applied over an area larger than the sensor. The sensor is placed in the middle of a weighing scale (initially an EK-200i from AND (A & D Company, Toshima City, Tokyo, Japan) as seen in Figure 2, later replaced with a Sartorius MC1 LC 4800 P (Sartorius AG, Goettingen, Germany)) and, with a known effective area of the sensor, the weight can be converted into pressure. The sensor is connected to the interrogator, and the scale is controlled by the computer using customised software developed using LabVIEW (National Instruments (Austin, TX, USA)).

When calibrating, it is necessary to define an ‘effective area’ of the sensor for the conversion of load weight to pressure. As described in Section 2.2, the sensor comprises an FBG encapsulated in a polymer, which then sits inside a textile cavity. The textile layer above the cavity acts like a membrane that exerts a force on the sensor and therefore the area of the top surface of the cavity (red dotted rectangle in Figure 1) was chosen as the effective area.

The process of loading and unloading is repeated three times for each sensor. The gradient values of the graph, which give values with a unit of nm/g, are then converted into pm/mmHg. A mean is then taken to be used as the sensitivity value for the sensor. 

#### 2.3.2. Strain Compensation

An experiment was designed to test the strain response of the sensor, and to measure the difference in strain sensitivity between the strain compensating FBG_ST_ and FBG_PST_ embedded within the polymer so that the strain compensation factor χ (Table 1) can be estimated. 

Strain was applied by hanging the fibre over a pulley and attaching a load—see Figure 3. The fibre jacket was held in place with magnetic clamps, with the fibre able to slide within the jacket. The polymer-encapsulated pressure sensing FBG was placed back into the mould in which it was made, thereby holding it in place and providing an anchor point. As a result, the applied strain will be experienced by the strain sensing FBG_ST_ and the pressure sensing FBG_PST_, but not by the temperature sensing FBG_T_. This will produce a value for the strain scaling factor introduced in Table 1.

Three fixed masses were applied to the fibre to produce the strain: 45 g, 73 g and 108 g.

#### 2.3.3. Blood Pressure Cuff

In Section 2.3.1 it is explained that using the loading arm for calibration requires the identification of an effective area of interaction for the applied force. To try and avoid the complications caused by trying to define this effective area, an alternative calibration method was designed, directly comparing the wavelength shift of the sensor to a known pressure applied using a blood pressure cuff. This also potentially overcomes measurement errors caused by the settling of the springs in the balance in the previous calibration method with the loading arm.

The experimental set-up is shown in Figure 4. An 80 mm diameter acrylic cylinder was enringed with a skin surrogate slab of 10 mm thickness (Polydimethylsiloxane (PDMS) saturated with polyester fibres to reach hardness of 00/70–00/86). The optical fibre pressure sensor is fixed onto the skin surrogate and a force is applied using a blood pressure cuff which in turn is connected to a syringe pump system delivering a total air volume of 110 mL at 200 mL/min. The syringe pump was implemented to deliver four consecutive infuse/withdrawal (I/W) cycles at 200 mL/min and the amount of pressure applied with the blood cuff against the sensor-skin system was measured in real time with a manometer (HD750, Extech, Nashua, NH, USA). The first I/W cycle of pumped air was used to better accommodate the cuff against the sensor/skin and, therefore, the last three cycles were used for calibration using the blood pressure cuff system. 

The optical fibre pressure sensor is connected to an interrogator (SmartScan, Smartfibres), which records the Bragg wavelength signal synchronously with the manometer in the same laptop.

### 2.4. Sensor Response 

A leg phantom was used to demonstrate the effects of a compression bandage on the sensor. Whilst, in the final suite of experiments, measurements were taken on a human subject using a methodology designed to mimic a real clinical situation.

#### 2.4.1. Phantom Leg 

A customised phantom limb was used to provide a stable and repeatable way to test the response of the sensor [35] (Figure 5). Although unable to recreate the movements and homeostatic properties of a living limb, it mimics the shape and texture of a human leg well enough to be a useful tool for modelling the behaviour of the sensor on a human subject. A location approximating the B1 position was identified for positioning the sensor. 

The testing procedure performed involved applying multiple layers of wrapping over the same position to demonstrate an increase in pressure. The phantom limb and sensor position are shown in Figure 5. The bandages used were Comprilan short stretch (BSN Medical, Hull, UK).

#### 2.4.2. Healthy Human Subject

The function of the optical fibre sensor was tested on B1 position of the lower leg of a healthy human subject, as shown in Figure 6. The sensor was attached to the leg and a period of 30 s was allowed to elapse to allow the sensor to adjust to the body temperature of the leg. This is necessary because the polymer encapsulating FBG_PST_ has slightly different thermal properties to the bare fibre, and so both take a slightly different amount of time to stabilise to the environmental temperature. At this point, the FOPS is manually calibrated to zero by setting the current wavelength value as 0 mmHg in the operating software.

Compression bandages (Putterbinde elastic bandage, Paul Hartmann AG, Heidenheim, Germany) were applied on the lower limb by a specialist (author A.P.) who has 32 years of clinical nursing experience. After application of the bandage, the human subject performed a series of motions including 30 s of supine, 30 s sitting, 30 s standing, 10 times calf raising, 30 s standing, 30 s sitting, and 30 s supine in sequence. This motion uses the principles outlined by medical experts and representatives of the compression bandaging industry [26] and has been used by other research groups working within this field [11]. 

This procedure was repeated ten times on the same human subject, with the same sensor reapplied and a fresh bandage used each time. As a comparison the pressure is also measured using the PicoPress during application of the compression bandage, with both sensors being placed at the approximate location of the B1 position on the human subject. This was done on every other wrap to determine whether the inflation of the bladder had any influence on the fibre optic pressure sensor (FOPS) measurement, meaning ten wraps in total with the FOPS but only five with the PicoPress in tandem as a comparison.

Permission for human subject studies was granted by the Faculty Research Ethics Committee for the Faculty of Engineering, in line with guidance at the University of Nottingham.

## 3. Results

In this section, the results are displayed for the calibration experiments outlined in Section 2 with interpretation of these results in the context of sub-bandage pressure measurement. Results are also shown to demonstrate the response of the FOPS to compression bandages being applied on a phantom limb, and finally pressure values produced in a mock clinical application.

### 3.1. Calibration 

#### 3.1.1. Calibration of the Pressure Sensor

The graph displayed in Figure 7 is a typical example of a calibration graph produced by the setup described in Section 2.3.1, giving a linear equation of *y* = 6.47 × 10^−4^*x* − 0.0031 (*y* being the wavelength shift in nanometers and *x* the mass in grams), and a coefficient of determination R^2^ = 0.9972. This process was repeated three times for this sensor, then the value converted from nm/g to pm/mmHg. This produces a sensitivity of 1.5 pm/mmHg for this sensor, with a standard deviation of 0.051 pm/mmHg. Across the ten sensors the mean sensitivity was 1.0 pm/mmHg with a standard deviation of 0.36 pm/mmHg (7.7 pm/kPa, standard deviation 2.72 pm/kPa). An example calibration graph for each sensor is provided in Appendix A.

The non-linearity at the low mass is thought to be due to the textile housing absorbing the initial load, and at the end is due to the loading arm coming to rest and the settling of the springs in the scale.

#### 3.1.2. Calibration Using Strain Rig

Figure 8a shows a typical response from the 3 FBGs to the strain caused by applying a load to the fibre with the application of the mass repeated three times for each mass using the experimental set up shown in Figure 2. As expected, FBG_T_ shows very little response as the distal end is not anchored and it cannot be elongated, so it is isolated from the effects of unwanted strain. However, the encapsulating polymer influences the strain response of FBG_PST_; only producing approximately half the wavelength shift seen in the bare fibre. This is compensated for within the pressure calculation algorithm, as shown in Table 1, by adjusting the strain compensation χ by a suitable amount before subtraction.

The relationship between the effect of strain on FBG_PST_ and FBG_ST_ is derived from the graph in Figure 8b. The mean value for the scaling factor for strain compensation between the two FBGs (Table 1) across all sensors was found to be χ = 2.22 with a standard deviation of 0.167, meaning that the FBG_ST_ outside the polymer is just over twice as sensitive to the effects of strain as FBG_PST_ inside the polymer.

#### 3.1.3. Calibration with Blood Pressure Cuff

Figure 9a shows the results of comparing the wavelength shifts of the FBGs to the pressure recorded by the manometer within the blood pressure cuff from the experimental set up in Figure 4. The FBG_PST_ sensor produces a wavelength shift that is in close agreement with the changes recorded by the manometer. The results also demonstrate that FBG_T_ has only a small response to pressure increase. FBG_ST_ also has a small response, indicating that the tubing and textile protect the fibre from strain when the cuff air pressure increases. 

The sensitivity value derived from the unloading cycles was always slightly lower than the value from the loading cycles, suggesting a slight hysteresis in the sensor. Loading and unloading graphs for the four cycles are shown in Figure 9b. This slight hysteresis is most noticeable at the low pressure values of the experiment; it is believed that the most likely cause is the textile housing maintaining a low pressure on the sensor even once the blood cuff has been deflated. Numerical analysis is provided using a method adapted from the Engineering Statistics Handbook (NIST/SEMATECH [36]) based on the difference between the values produced by the upscale and downscale readings. The largest difference in the wavelength shift during loading and unloading for identical pressures is displayed on the plot, with the largest disagreement being a shift of 6.1 pm.

Although a slight hindrance for calibration, the sub-bandage pressure application for which the sensor is designed (sub-bandage pressure) would mean that only the values under loading (when the bandage is in place) are of interest.

The mean sensitivity produced across 20 cycles of loading and unloading by this calibration method was 1.1 pm/mmHg with a standard deviation of 0.064 pm/mmHg (Appendix A).

### 3.2. Demonstrating Sensor Response

To create a scenario where the pressure exerted by the compression bandage was increased, the bandage was wrapped around the same location on the phantom leg, where the sensor was positioned, multiple times. The increasing pressure steps as each layer of bandage is added can be seen quite clearly on the left-hand side of the graph in Figure 10. The decreasing pressure steps as the layers are removed are less clear, which is believed to be due to a slightly delayed response caused by the textile housing for the sensor, maintaining some of the pressure exerted by the bandage. A similar explanation is given for the failure to return to zero after the bandage is removed.

Each plateau shows an initial high value and then a gradual relaxation caused by the wrapping process during which the bandage is applied firmly over the sensor but then gradually relaxes as the bandage is moved to the opposite side of the leg. There will also be some relaxation in the bandage material.

### 3.3. Sensor Tests with Bandages on Healthy Human Subjects

A typical test result of the FBG-based pressure sensor on a single healthy subject is shown in Figure 11 alongside the pressure measured using the PicoPress. Other examples are included in the Appendix A, taken with another four versions of the FOPS. Appendix A shows two sets of wrapping data with the same sensor taken one after the other.

The baseline pressure before the bandage was applied was set to be 0 mmHg. The volunteer carries out the motions as described in Section 2.4.2 and illustrated on Figure 11, with both sensors attached, meaning that although both sensors are at the approximate B1 position, they are not measuring at precisely the same location.

In Figure 11, the wrapping of the leg starts at 20–30 s, after setting the pressure value to 0 mmHg as described previously in Section 2.4.2. The steady rise in the pressure value of the FOPS at this time is due to the body temperature of the nurse (author A.P) applying the bandage; there is a slight delay in temperature compensation due to the polymer capsule having different thermal properties to the bare fibre, essentially providing a layer of insulation around the FBG. Between 50–70 s, the bandage is applied over the B1 position where the two sensors are placed, as shown by the significant increase in the pressure values of both sensors. The spikes in the FOPS values are seen up to around 90 s are movement artefacts caused by the bandage being applied.

The steady rise in the FOPS value between 100 and 130 s is not fully understood and is not present on all the traces. This may be due to the human subject shifting and relaxing their leg once the nurse has finished applying the bandage; it may require further temperature adjustments as the nurse steps away and the bandage adjusts to the body temperature of the human subject with different temperature rates for FBG_PST_ and FBG_ST_ (caused by FBG_PST_ being encapsulated in a polymer whereas FBG_ST_ is bare fibre). It may also be due to homeostatic processes within the body as it adjusts to the pressure being applied by the bandage; increased blood flow to compensate for the restriction of the blood vessels, for example.

The leg is rested supine on a stool until 130 s (box 1 on Figure 11), when it is then moved and the human subject places their foot on the floor (box 2 on Figure 11). The PicoPress does not record a change in value during this motion, but the FOPS commonly shows a drop in pressure. At around 160 s, the human subject stands (box 3 on Figure 11), causing a pressure rise in both sensors and a varying pressure value as the calf flexes and relaxes to maintain balance.

The ‘noisy’ section of signal at 200 s is due to the human subject carrying out 10 calf raises (box 4 on Figure 11). The patient then stands still after completing the calf raises (box 5 on Figure 11). Both sensors commonly show a slightly different pressure after the calf raises, possibly due to movement of the bandage or a slight increase in blood to the calf muscle. At 250 s, the human subject sits down, shown commonly as a pressure drop in both sensors (box 6 on Figure 11).

At approximately 280 s, the human subject returns their leg to a supine position by resting their foot on a stool (box 7 on Figure 11). Both sensors show an increase in pressure at this point. The spike in the FOPS at around 320 s, followed by a rapid decrease in the pressure values on both sensors, is the bandage being removed.

The FOPS value taking longer to return to zero is believed to be due to the textile housing for the sensor maintaining some pressure on the polymer capsule. That the value drifts below zero perhaps indicates that more time was needed to adjust for temperature at the beginning of the recording.

#### Bland–Altman Analysis

To produce a Bland–Altman plot, the five traces directly comparing the response of the FOPS and the PicoPress were divided into the seven sections associated with the movements described in Section 2.4.2 and shown in Figure 11 (other datasets are included in Appendix A). The data from sections applying and removing the bandage were removed, and small gaps were left between each section to attempt to remove the periods where the human subject was transitioning between positions. 

The results for each of the seven sections from each of the five datasets were then plotted as a Bland–Altman plot; comparing the bias between the sensors (FOPS value minus PicoPress value) to the mean value measured. The bias values from the 5 plots were then combined to produce a median bias and LOA values for each of the seven sections; so, each of the seven sets of values in the table below were produced from 5 Bland–Altman plots. The results of this process are shown in Table 2.

## 4. Discussion

The FBG-based pressure sensor shows distinctive pressure changes for different motions, indicating its potential use to help train tissue viability nurses; regulate bandage application to achieve desirable pressure levels; support the treatment of venous leg ulcers and lymphoedema; and evaluate the performance of compression bandages. Since multiple FBGs can be easily fabricated in one fibre and distinguished through wavelength division multiplexing techniques, the sensor can be further developed to monitor the pressure distribution along the lower leg under the whole of the bandaged area.

Throughout this paper, all the results displayed are from one sensor, but overall, ten sensors were produced to investigate variability in the manufacturing process. Although the FBGs and the textiles were produced commercially with a high degree of repeatability, the polymer encapsulation of the pressure measurement of FBG was made by hand in the lab, which led to variations in sensitivity between the ten sensors. These can be individually calibrated, which resulted in a range of sensitivity values from 0.4–1.5 pm/mmHg (Appendix A). This manual production technique led to some of the sensors having a ‘preferred orientation’, for which way up they were positioned on the human subject, as the fibre did not run directly through the vertical centre of the capsule. This can be improved by using an automated manufacturing process, such as injection moulding of the polymer capsule.

During sensor development, it was found that the FBG-based sensor can show negative pressure values if it is taped on the skin directly, which is associated with the polymer being compressed in the axial direction. This is likely to be due to the formation of an indention at the location of the sensor on soft skin and the compression of the bandage around the polymer. To avoid this effect, the sensor is enclosed inside a textile housing, as shown in Figure 1, which prevents the bandage from wrapping around the polymer or the polymer sinking into an indentation in the skin.

Another common issue during calibration and experimentation was the failure of the sensor to return to zero when the bandages were removed (Figure 11). Although the value always returns to near zero, it often sits above or below the initial zero point. As described previously (Section 2.4.2), the zero-pressure value is calibrated manually after the sensor has been attached to the human subject or phantom but before the bandage is applied. Reasons for this ‘return to zero’ error could be a result of the textile housing maintaining some pressure on the sensor after the bandaging has been removed, or from a hysteresis of the polymer itself. Some improvements were made by pre-compressing the textiles before using them to house the sensor. This was achieved by keeping the textile under a large metal plate for 24 h before use.

Another consideration for the return to zero error was whether the temperature was having an effect, as spikes could often be seen when handling the sensor despite a method being in place for temperature compensation. This was explained as being the result of a slight delay in the compensation method due to the polymer being slower to adjust to temperature changes than the bare fibre. This effect is not significant over the time period of the measurement, as a 30 s thermal equilibrium interval was added to the user procedure.

There are alternative methods for compensating for temperature within an FBG-based sensing system. One such example uses the thermal properties of the coating surrounding the FBG to model the response to temperature [30], whilst another method recently demonstrated shows that the FBG sensor responds more quickly to temperature changes than to pressure changes, meaning that a frequency analysis of the changing signal can be used to separate the two effects [37]. Both methods are designed to be used in systems with a much greater change of temperature than experienced here, however, where the sensor is largely kept at body temperature. As a result, the changes in temperature experienced in this investigation may be too small for these alternative methods. 

During calibration, as described in Section 2.3.1, it was necessary to define an ‘effective area’ of the sensor for the conversion of load weight to pressure. As described in Section 2.2, the sensor comprises an FBG encapsulated in a polymer which then sits inside a cavity within the textile housing. The textile above the cavity acts like a membrane that exerts a force on the sensor; therefore, the area of the top surface of the cavity was chosen as the effective area. This is an aspect of this work that requires further investigation, as an argument can also be made for other values to act as the effective area; for example, the surface area of the top of the polymer capsule.

An alternative calibration method was attempted using a blood pressure cuff. The average sensitivity produced by this method was 1.1 pm/mmHg, compared to a mean sensitivity of 1.0 pm/mmHg from the calibration using the scale and loading arm. The wrapping data included in this paper uses the loading arm calibration method for determining the sensitivity of the FOPS. This value was used to reverse-engineer a value for the effective area for the loading arm calibration method, giving an area of 1.32 × 10^−4^ m^2^. The area assumed in the previous calibration method was 1.65 × 10^−4^ m^2^. This disagreement also shows that there is further work to be carried out in determining the most appropriate calibration method for the sensor.

When compared to a commercially available sensor, the FOPS consistently recorded values below the pressure value measured by the PicoPress, as seen in Table 2. This is not necessarily an error with the FOPS sensor, and the consistency of the variation between the two sensors suggests a systematic disagreement in the way the pressure is measured. Although both devices produce pressure values, they measure force, which is a vector quantity meaning that the direction of the applied force is important. The PicoPress measures pressure by using an air-filled bladder which, due to its curved shape, will be susceptible to forces applied to the sensor from many directions. The FOPS is designed to only measure the force being applied perpendicular to the orientation of the sensor, and so will experience a lesser overall applied force than a pneumatic sensor. This proposition is further evidenced by the data showing that the disagreement between the sensor is minimum when the human subject’s leg is supine; at this point, the bandage will be most smoothly stretched along the leg. As soon as the human subject lowers their leg, the bandage will undergo some deformation (e.g., bunching of the bandage layers) and the opportunity arises for lateral pressure to be applied to the PicoPress, from which the FOPS will be isolated.

Other researchers in this field have produced an FBG-based pressure sensor with a sensitivity of 4.8 pm/mmHg [11], as compared to our sensors with a peak sensitivity of 1.5 pm/mmHg. This also shows there are potential improvements to be made to the design of our sensor. For example, encapsulating the pressure sensing FBG with a softer or different shaped polymer.

## 5. Conclusions

In summary, a skin interface pressure sensor based on FBGs has been developed to measure the absolute sub-bandage pressure. The pressure sensor can measure the pressure in the range of 0–360 mmHg with a mean sensitivity of 1.0 pm/mmHg (Standard deviation 0.36 pm/mmHg). Pressure measurements on a custom produced phantom leg show the sensor can produce a stable reading that responds to changing pressures applied by a compression bandage. Measurements on human subjects during bandaging performed by an experienced nurse demonstrated that the pressure sensor can be used to indicate changes in pressure upon application of the bandage and performing different motions (sitting, standing, calf raising). The developed sensor has the potential advantage of being able to take a series of measurements along the leg under the whole of the bandaged area using a single fibre and would cause less damage to fragile skin thanks to the use of a textile housing and a thin optical fibre to deliver signals.

## Figures and Tables

**Figure 1 sensors-22-01798-f001:**
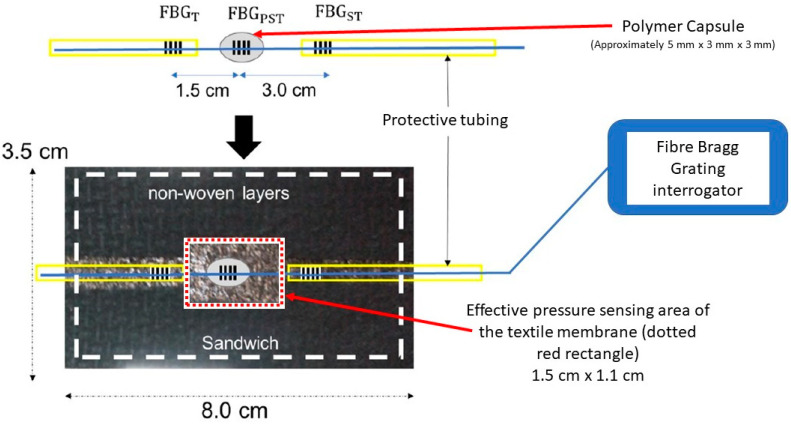
Schematic of the optical fibre-based pressure sensor. An FBG (FBG_PST_) is encapsulated in a polymer to transduce applied pressure into a measurable strain–effective pressure sensing area shown in a red rectangle. Two reference FBGs (FBG_ST_ and FBG_T_) sensitive to strain and temperature respectively are used to compensate for these effects on the pressure measurements. The sensors are enclosed in a textile ‘sandwich’ and are connected to an interrogator unit via an optical fibre patch cord.

**Figure 2 sensors-22-01798-f002:**
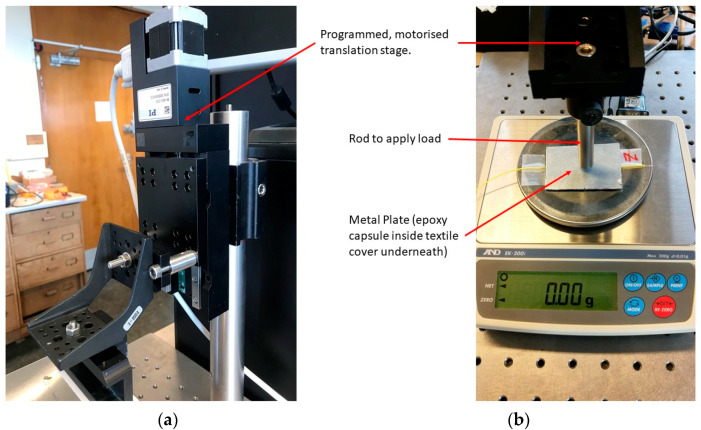
Customised calibration setup. (**a**) The motorised translation stage is programmed to lower the rod onto the weighing scale. The pressure sensor in its textile housing is attached to the plate of the weighing scale to keep it in place during the calibration. (**b**) The load is transferred from the translation stage to the pressure sensor via a metal rod and an aluminium metal plate. The probe is lined up over the central FBG of the sensor before starting the loading process.

**Figure 3 sensors-22-01798-f003:**
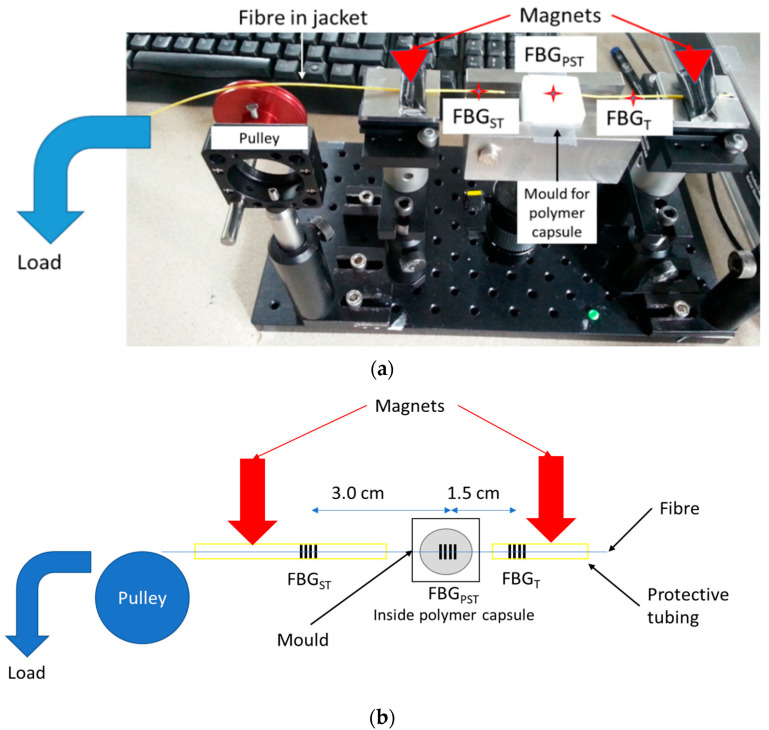
Experimental setup for testing the strain response of the fibre. The mould provides an anchor point to isolate FBG_T_ from the effects of strain, the magnets hold the yellow protective tubing in place and the rest of the fibre can slide within the yellow jacket. (**a**) Labelled photograph; (**b**) labelled schematic.

**Figure 4 sensors-22-01798-f004:**
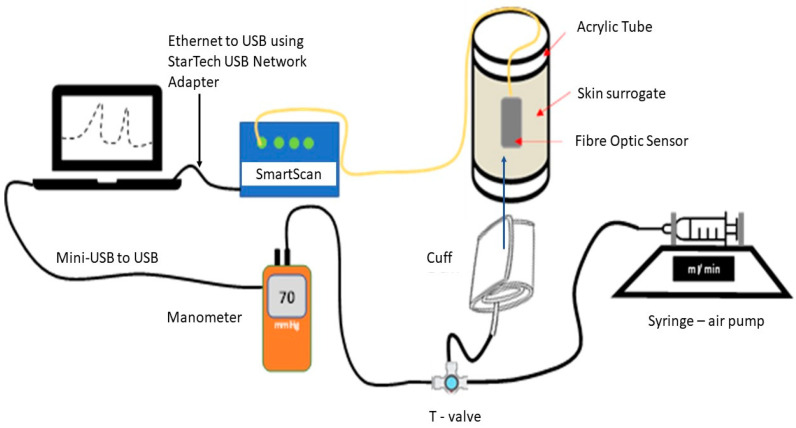
Experimental setup for measurements using a blood pressure cuff. A controlled volume of air is added via syringe pump to the cuff, which is wrapped around the materials on the acrylic tube. The pressure of the cuff is recorded by the manometer and is compared to the wavelength shift recorded by the interrogator.

**Figure 5 sensors-22-01798-f005:**
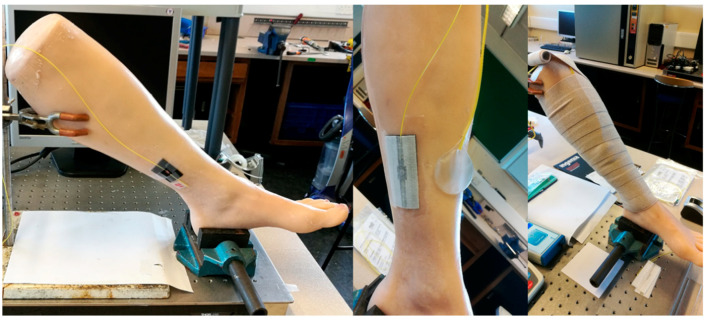
Custom-made phantom leg shown with sensors (optical fibre and PicoPress) at the approximate B1 position and with a full bandage wrap.

**Figure 6 sensors-22-01798-f006:**
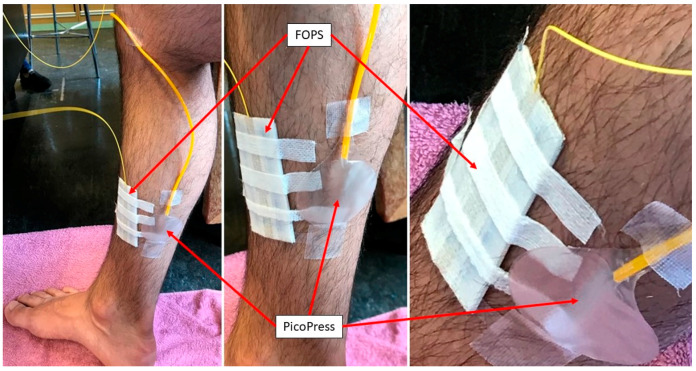
Photos of the optical fibre sensor and the PicoPress at approximately the B1 position on a human subject.

**Figure 7 sensors-22-01798-f007:**
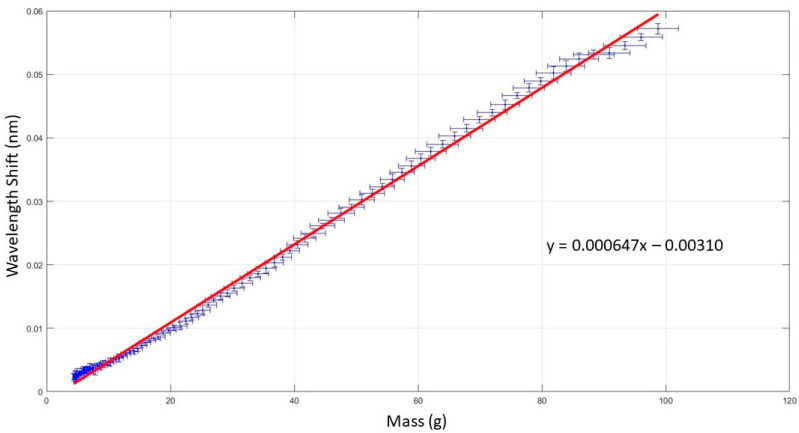
Typical calibration results of the FOPS: wavelength shift against load weight using experimental set up illustrated in Figure 2.

**Figure 8 sensors-22-01798-f008:**
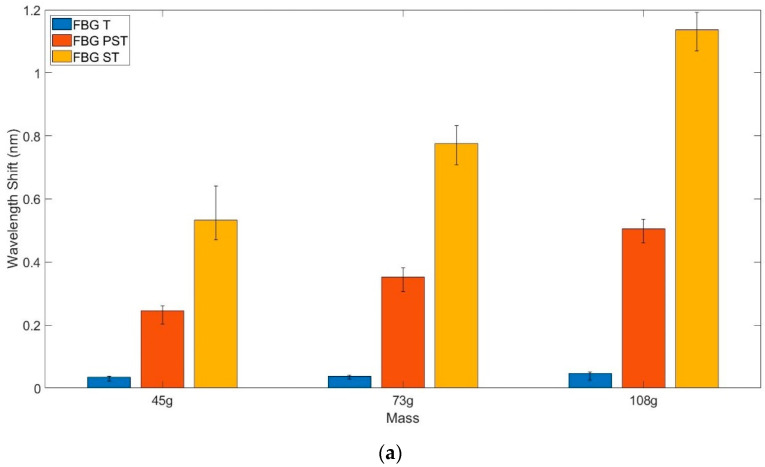
(**a**) Typical response of the 3 FBGs to the load applied for strain compensation measurements (using experimental set up in Figure 3), showing the mean wavelength shift for each applied mass. (**b**) Typical results showing the relative wavelength shift in response to strain from the FBG_PST_ encapsulated with polymer and FBG_ST_ in the bare fibre. Although only three different masses were used, the sampling rate of the sensor compared to the settling time of the scales means a range of wavelengths were recorded for each mass.

**Figure 9 sensors-22-01798-f009:**
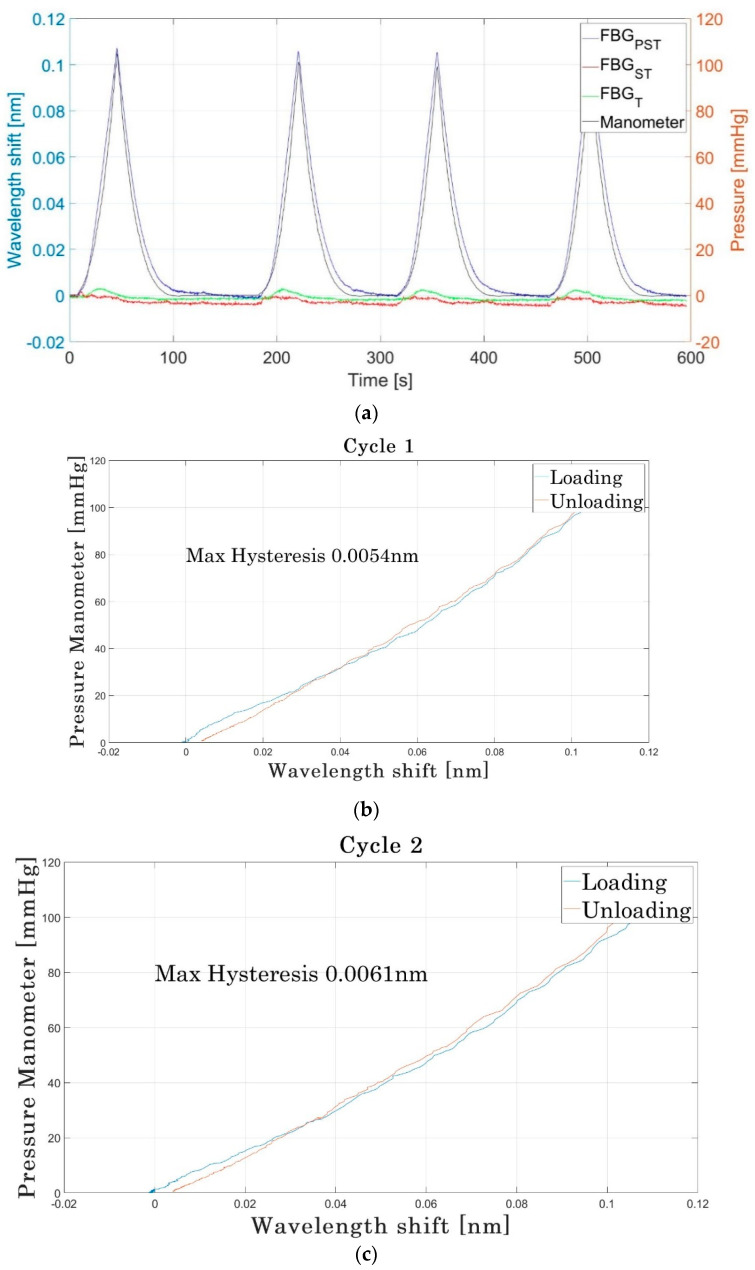
(**a**) Typical results: comparison of wavelength shift produced by the sensor to the pressure from a manometer, taken on a rigid cylindrical tube and controlled by a blood pressure cuff (using the experimental set up in Figure 4). (**b**–**e**) Pressure against wavelength shift for four loading and unloading cycles.

**Figure 10 sensors-22-01798-f010:**
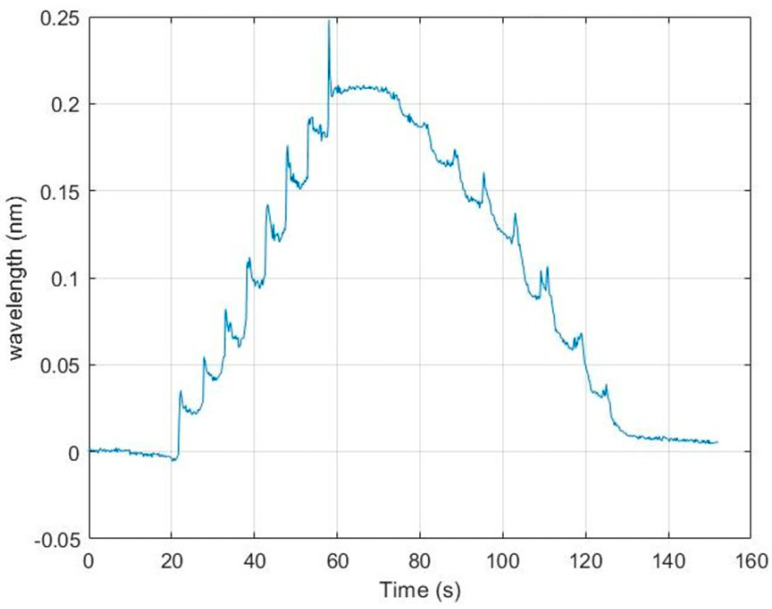
Typical results: wavelength shift against time underneath a compression bandage wrapped multiple times (in this case 8 layers) on a single location on the phantom leg. The displayed wavelength values are strain and temperature compensated.

**Figure 11 sensors-22-01798-f011:**
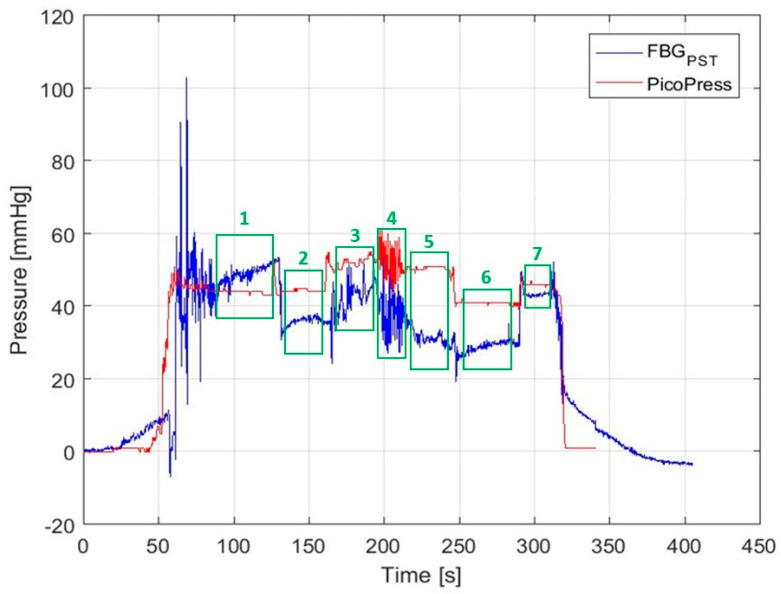
Pressure recorded by the Fibre Optic Sensor (after compensation for strain) compared to the PicoPress as shown in the experimental set up in Figure 6. The trace has been divided into 7 sections showing the different activities the subject performs during the wrapping experiment, which are then used to perform a Bland–Altman assessment of the data across all five wrapping experiments. 1: Sitting; leg supine; 2: Sitting; leg down; 3: Standing; 4: Calf raises; 5: Standing; 6: Sitting, leg down; 7: Sitting, leg supine.

**Table 1 sensors-22-01798-t001:** Parameters monitored and the corresponding FBG measured (used in in conjunction with Equations (3) and (4)).

Parameter	Measurement
Temperature	FBG_T_
Strain	FBG_ST_ − FBG_T_
Pressure	FBG_PST_ − [FBG_ST_ × χ]

Note χ is the strain scaling factor is explained in Section 2.3.2 and Section 3.1.2.

**Table 2 sensors-22-01798-t002:** Data from the five wraps directly comparing the FOPS to the PicoPress. Each wrap was broken down into sections based on the seven different movements performed (as shown in Figure 11) and a Bland–Altman plot was produced from each of these sections. Each of the Bias values is therefore the median of five different Bland–Altman plots. LOA–limit of agreement between PicoPress and FOPS. LOA1 is the upper limit and LOA2 is the lower limits of agreement i.e., 1.96 × standard deviation.

Section: Action	Median Bias and Range (mmHg) (*n* = 5)	Median LOA1 and Range (mmHg) (*n* = 5)	Median LOA2 and Range (mmHg) (*n* = 5)
1: Sitting, leg supine	−4.64 (−27.5–9.00)	−1.59 (−20.9–10.5)	−7.67 (−34.1–0.678)
2: Sitting, leg down	−16.8 (−36.0–3.57)	−10.5 (−30.8–7.54)	−23.2 (−41.1–−0.385)
3: Standing	−20.4 (−32.5–−6.55)	−9.71 (−24.7–0.278)	−31.1 (−42.1–−13.4)
4: Calf raises	−17.1 (−27.4–−7.62)	−1.72 (−12.6–4.81)	−32.4 (−42.0–−20.0)
5: Standing	−17.1 (−30.1–2.00)	−7.08 (−24.2–7.45)	−22.9 (−36.0–−3.42)
6: Sitting, leg down	−17.9 (−29.0–−0.496)	−13.5 (−26.0–2.39)	−22.2 (−32.0–−3.37)
7: Sitting, leg supine	−4.87 (−14.6–5.33)	2.66 (−8.53–9.44)	−16.6 (−18.1–1.24)

## Data Availability

The data presented in this study are available on request from the corresponding author.

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
