# Peer review of "Fibre Bragg Grating Based Interface Pressure Sensor for Compression Therapy"

_sensors, 2022, doi:10.3390/s22051798_

Round 1
Reviewer 1 Report
In the paper “Fibre Bragg grating based interface pressure sensor for compression therapy”, the authors presented the characterization process of the skin interface pressure sensor based on fibre Bragg gratings (FBGs) to measure the absolute sub-bandage pressure. The sensor device consists of a three femtosecond FBGs. In order to increase the pressure sensitivity, the FBGs were encased in a polymer (UV curing adhesive).
The sensing process is based on when a load is applied the polymer deforms and transfers the load to the fiber, such as a deformation that changes the FBG.
Sensors based on FBGs, such as pressure and strain sensors for health area, are not a new subject. However, the novelty of the article is the new configuration of the optical sensor based on FBG for application in compression therapy.
The article presents characterization assays and also measurements on a human subject during movement to simulate a clinical application.
The paper is interesting and has application potential for medical and biochemical fields.
In general, the manuscript is clear and well written. However, there are some questions that could improve the paper. Please authors, consider the following comments/suggestions:
- Could the authors explain why they used femtosecond FBGs instead of commercial FBGs? What are the advantages of femtosecond FBGs in biomechanical or biomedical applications?
- In figure 9b the authors presented the sensitivity graph, however, it is not possible to notice the hysteresis, because the graphs are superimposed. Could the authors to adjust curves for each graph and then compare the coefficients of the adjustments for hysteresis analyses?
- The authors reported on page 13: “This slight hysteresis was assumed to be due to the textile housing”. Could the authors report the time between one assay and another (loading and unloading cycles)? Could a longer time interval resolve this hysteresis?
- Could the authors explain the pressure steps and the spikes in the section 3.2 “Demonstrating sensor response”?
- Some figures are not in the proper size.
- Some typographical errors (for example in line 94) and reference errors (for example in [20]) in the text need attention.
- Also, is necessary to revise some units (for example, in sometimes is used picometres per mmHg and in others is used pm/mmHg). Some symbols equations need to be standardized (for example, in sometimes is used asterisk for multiplication sign (FBGST * χ) and in others is used the X).
After the minor revisions, the manuscript should be suitable for publication on Sensors Journal.
Reviewer 2 Report
This paper, interesting piece of work, reports a new configuration of an FBG based optical sensor with application in compression therapy. The sensitivity of the FBG to pressure is increased by embedding the FBG inside a polymer. I have some comments to highlight the impact of this paper.
- Introduction: I need introduction of some recent paper about the diaphragm embedded material on FBG to get more sensitivity and insensitive to temperature. Please add and read: Optics express 26 (16), 20590-20602, 2018.
- How about the use of the polymer Vitralit 1655 (Panacol Adhesives, Frankfurt, Germany) which is a UV curing adhesive, comparing with the related in Sensors 20 (11), 3026, 2020. Please add some words.
- There are some literature where just one FBG can measure pressure insensivity to temperature (Optics & Laser Technology 112, 77-84, 2019) and it needs to be consider in the discussion.
- How about the reproducibility of the results usind identical probes?
Round 2
Reviewer 2 Report
The paper is ready for publication with all comments well addressed.